# Resilient SARS-CoV-2 diagnostics workflows including viral heat inactivation

**Maria Jose Lista**[1,2©], **Pedro M. Matos**[1,2©], **Thomas J. A. Maguire**[1,3©], **Kate Poulton**[1,2©], **Elena Ortiz-Zapater**[1,4,5], **Robert Page**[1,6], **Helin Sertkaya**[1,2], **Ana M. Ortega-Prieto**[2], **Edward Scourfield**[2], **Aoife M. O'Byrne**[1,7], **Clement Bouton**[1,2], **Ruth E. Dickenson**[1,2], **Mattia Ficarelli**[1,2], **Jose M. Jimenez-Guardeño**[2], **Mark Howard**[1,5], **Gilberto Betancor**[1,2], **Rui Pedro Galao**[1,2], **Suzanne Pickering**[1,2], **Adrian W. Signell**[1,2], **Harry Wilson**[1,2], **Penelope Cliff**[8], **Mark Tan Kia Ik**[9], **Amita Patel**[9], **Eithne MacMahon**[9], **Emma Cunningham**[9], **Katie Doores**[1,2], **Monica Agromayor**[1,2], **Juan Martin-Serrano**[1,2], **Esperanza Perucha**[1,7], **Hannah E. Mischo**[1,2], **Manu Shankar-Hari**[1,2], **Rahul Batra**[9], **Jonathan Edgeworth**[9], **Mark Zuckerman**[1,10], **Michael H. Malim**[1,2], **Stuart Neil**[1,2], **Rocio Teresa Martinez-Nunez**[1,2]\*

**1** King's College London Diagnostics Team at Guy's Campus, London, United Kingdom, **2** Dept. Infectious Diseases, School of Immunology and Microbial Sciences, King's College London, London, United Kingdom, **3** Dept. Inflammation Biology, School of Immunology and Microbial Sciences, Asthma UK Centre in Allergic Mechanisms of Asthma, King's College London, London, United Kingdom, **4** Randall Centre for Cell & Molecular Biophysics, King's College London, London, United Kingdom, **5** Peter Gorer Department of Immunobiology, King's College London, London, United Kingdom, **6** King's Health Partners Integrated Cancer Centre, School of Cancer and Pharmaceutical Sciences, Guy's Hospital, King's College London, London, United Kingdom, **7** Centre for Inflammation Biology and Cancer Immunology (CIBCI), Centre for Rheumatic Diseases (CRD–EULAR Centre of Excellence), King's College London, London, United Kingdom, **8** Viapath pathology laboratories at St Thomas' Hospital, London, United Kingdom, **9** Centre for Infectious Diseases Research, St Thomas' Hospital, London, United Kingdom, **10** South London Specialist Virology Centre, King's College Hospital, London, United Kingdom

© These authors contributed equally to this work.

\* rocio.martinez_nunez@kcl.ac.uk

**Data Availability Statement:** All relevant data are within the manuscript and its Supporting Information files. Additional files are provided on

## Abstract

There is a worldwide need for reagents to perform SARS-CoV-2 detection. Some laboratories have implemented kit-free protocols, but many others do not have the capacity to develop these and/or perform manual processing. We provide multiple workflows for SARS-CoV-2 nucleic acid detection in clinical samples by comparing several commercially available RNA extraction methods: QIAamp Viral RNA Mini Kit (QIAgen), RNAdvance Blood/Viral (Beckman) and Mag-Bind Viral DNA/RNA 96 Kit (Omega Bio-tek). We also compared One-step RT-qPCR reagents: TaqMan Fast Virus 1-Step Master Mix (FastVirus, Thermo-Fisher Scientific), qPCRBIO Probe 1-Step Go Lo-ROX (PCR Biosystems) and Luna® Universal Probe One-Step RT-qPCR Kit (Luna, NEB). We used primer-probes that detect viral N (EUA CDC) and RdRP. RNA extraction methods provided similar results, with Beckman performing better with our primer-probe combinations. Luna proved most sensitive although overall the three reagents did not show significant differences. N detection was more reliable than that of RdRP, particularly in samples with low viral titres. Importantly, we demonstrated that heat treatment of nasopharyngeal swabs at 70˚C for 10 or 30 min, or 90˚C for 10 or 30 min (both original variant and B 1.1.7) inactivated SARS-CoV-2 employing plaque assays, and had minimal impact on the sensitivity of the qPCR in clinical samples. These findings

the Open Science Framework (https://osf.io/uebvj/).

**Funding:** This work was funded by; King's Together Rapid COVID-19 Call awards to RTM-N, MHM, KD, SN, JE and MS-H, MRC Discovery Award MC/PC/15068 to SN, KD and MHM, a Huo Family Foundation Award to MHM, KD, RTM-N, SN, JE and MS-H, MRC Programme Grant MR/S023747/1 to MHM, Wellcome Trust Investigator Award 106223/Z/14/Z to MHM, NIAID Awards U54AI150472 and R01AI076119 to MHM. A.S., M.F. and R.E.D. were supported by the MRC-KCL Doctoral Training Partnership in Biomedical Sciences (MR/N013700/1); E. O-Z was supported by MR/S009191/1 to Parsons M and Santis G; M.F. was supported by the MRC (MR/R50225X/1); H.S was supported by the BBSRC (BB/P504609/1); R. P. was supported by the by the (NIHR) Biomedical Research Centre based at Guy's and St Thomas' NHS Foundation Trust and King's College London; T.J.A.M. was supported by Asthma UK at the Asthma UK Centre in Allergic Mechanisms of Asthma; P.M., S. P. and H.W. were supported by Wellcome Trust WT098049AIA to Neil S. and Swanson C.; M.J.L.B and C.B. were supported by MRC (MR/S000844/1) to Neil S. and Swanson C. This UK funded award is part of the EDCTP2 programme supported by the European Union; K.P. was funded by KHP Challenge Fund to R.T.M-N; M. H. was funded by RP_007_20190305 from Kidney research UK; H.M. was funded by the Wellcome Trust and Royal Society Sir Henry Dale Fellowship 218537/Z/19/Z. Links to funders website: MRC: https://mrc.ukri.org/ Wellcome Trust: https://wellcome.org/ King's Health Partners: https://www.kingshealthpartners.org/ Asthma UK: https://www.asthma-allergy.ac.uk/ King's Together: https://www.kcl.ac.uk/research/funding-opportunities/seedfund Huo Family Foundation: https://huofamilyfoundation.org/ NIAID: https://www.niaid.nih.gov/ KCL-MRC: https://kcl-mrcdtp.com/ BBSRC: https://bbsrc.ukri.org/ NIHR-GSTT: https://www.guysandstthomasbrc.nihr.ac.uk/ EDCTP2: https://www.edctp.org/ Kidney Research UK: https://kidneyresearchuk.org/ The funders had no role in study design, data collection and analysis, decision to publish, or preparation of the manuscript.

**Competing interests:** The authors have declared that no competing interests exist.

make SARS-CoV-2 testing portable in settings that do not have CL-3 facilities. In summary, we provide several testing pipelines that can be easily implemented in other laboratories and have made all our protocols and SOPs freely available at https://osf.io/uebvj/.

## Introduction

"Test, test, test"–this was the message from the World Health Organization's (WHO) Head Tedros Adhanom Ghebreyesus on the 16[th] of March 2020. This message is still current, more than a year after the pandemic was declared. To fight the exponential spread of SARS-CoV-2, measures of social distancing have been imposed in many countries worldwide, while others are now in a phase of de-escalation. Social distancing and lockdown measures have resulted in stagnant or dropping numbers of new infections. However, appearance of outbreaks has proven inevitable in places where measures have been relaxed. COVID-19 immunisation has decreased transmission in certain countries; however, these are few nations and the spread of new variants makes testing as important as before. Test, Trace and Isolate have been essential to halt SARS-CoV-2 infection. Non-PCR tests such as lateral flow tests have proven useful particularly in the case of symptomatic testing [1]; however, large scale PCR-based testing is essential to contain and prevent outbreaks due to its high sensitivity. This is particularly relevant in asymptomatic individuals and should also be central in implementing an 'exit strategy' plan.

In order to increase testing capacity, many countries rely on centralised efforts to build large diagnostic centres. However, the involvement of smaller academic or commercial laboratories has proven helpful and necessary too [2–5]. These decentralised laboratories can repurpose existing equipment and technical expertise for SARS-CoV-2 testing, for example by comparing methods of extraction vs extraction-free methods or samples treated with heat [6–11], combining heat with proteinase K treatments to improve sensitivity [12] or establishing sensitivity of primer-probe pairs [13,14]. The UK government document "Guidance for organisations to seek supporting the COVID-19 testing programme" published on the 9[th] of April 2020, by the Department of Health and Social Care, clearly welcomed academic institutions to increase testing capacities within the UK, referred to here as NHS-helper labs. However, due to global high demand of the kits and reagents used in the WHO, CDC (Centres for Disease Control, US), ECDC (European Centre for Disease Prevention and Control) and PHE (Public Health England) ratified testing strategies, the NHS-helper labs were encouraged to use alternative strategies that would not interfere with the reagent demand of larger testing facilities. Moreover, helper laboratories could provide their research expertise and experimental validation of other kits enabling clinical labs to benefit from their results. We set out to perform this task.

Here we describe different strategies for SARS-CoV-2 PCR-based detection by employing reagents that are not currently used in the NHS setting. We also performed heat inactivation employing dry bead baths of SARS-CoV-2, for both the original variant and the more recent alpha (B 1.1.7) variant and present data supporting good limit of detection (LoD) for both variants after heat treatment. Within the UK, the NHS agrees on the use of alternative RNA isolation and qPCR protocols, providing these have been internally validated and discussed with the local NHS partner. To increase visibility of these alternative strategies, we have created a webpage under the Open Science Framework platform (https://osf.io/uebvj/) that we hope will stimulate exchange between smaller laboratory facilities, increase confidence in tested

alternative routes of RNA isolation and viral RNA detection and thereby expedite the establishment of smaller academic testing centres.

## Materials and methods

All materials with their catalogue numbers are available at https://osf.io/uebvj/.

### Heat inactivation

Swab tubes containing Viral Transport Medium (VTM) were checked for cracks to ensure no viral material had leaked. Swab tubes were then transferred to a water bath (Grant) containing dry metallic beads (Starlab) preheated to 70˚C or 90˚C, ensuring the entire swab tube (including lid) was covered by the beads. Samples were incubated in the following conditions: 70˚C for 10 mins, 70˚C for 30 mins, 90˚C for 10 mins, or 90˚C for 30 mins, then transferred back to Class I MSC and allowed to cool to room temperature prior to RNA extraction.

### RNA extraction

Serial dilutions were done in Hank balanced salt solution (HBBS) and 1% Bovine Serum Albumin (BSA) to closely mimic viral transport media.

**Qiagen QIAamp Viral RNA Mini Kit.** From swab tube, 140 µl sample was transferred to 1.5 mL screw-cap microcentrifuge tube and treated with 560 µl AVL, containing carrier RNA, followed by 560 µl molecular-grade 100% ethanol (Fisher Scientific). Samples were then taken out of the Class I MSC and CL-3 lab as AVL is known to inactivate SARS-CoV-2, transferred into QIAamp mini spin columns (Qiagen) and centrifuged according to manufacturer's instructions. Two wash steps were performed, with 714 µl buffer AW1 and 714 µl buffer AW2 (both Qiagen). RNA was then eluted from the columns with 40 µl RNase-free water (Ambion), followed by a second 40 µl elution to maximise RNA yield and giving a final RNA sample volume of 80 µl.

**Beckman Coulter Agencourt RNAdvance Blood Total RNA Kit.** Reagents were prepared prior to RNA extraction according to manufacturer's instructions. The protocol was conducted in a Class I MSC in a CL-3 lab. From a swab tube, 140 µl were transferred to a Zymo-Spin I-96 Plate (Zymo Research). 7 µl of Proteinase K/PK buffer and 105 µl of Lysis buffer was added to each sample, mixed and incubated at room temperature for 15 minutes. Following incubation, 143 µl of Bind1/Isopropanol was added to each sample, mixed, and the samples were left to incubate at room temperature for 5 min. The Zymo-Spin I-96 Plate was placed on ZR-96 MagStand (Zymo Research), and the magnetic beads left to form a pellet. The supernatant was removed, and the magnetic beads washed three times, first, with 280 µl of Wash buffer (Beckman Coulter), followed by two washes with 70% ethanol. Following the wash steps, RNA was eluted from the columns with 80 µl RNase-free water (Ambion).

**Omega Bio-tek Mag-Bind® Viral DNA/RNA kit.** Reagents were prepared prior to RNA extraction according to manufacturer's instructions. The protocol was conducted in a Class I MSC in a CL-3 lab. From a swab tube, 140 µl sample was transferred to a Zymo-Spin I-96 Plate (Zymo Research). 369.5 µl of Lysis mastermix was added to each sample, mixed, and incubated at room temperature for 10 minutes. Following incubation, 7 µl of Mag-Bind® Particles CNR and 7 µl of Proteinase K solution was added to each sample, mixed and incubated at room temperature for 5 minutes. The Zymo-Spin I-96 Plate was placed on ZR-96 MagStand (Zymo Research), and the magnetic beads left to form a pellet. The supernatant was removed, and the magnetic beads washed three times, first, with 280 µl of VHB buffer (Omega Bio-tek), followed by two washes with 350 µl SPR Wash Buffer (Omega Bio-tek). Following the wash steps, RNA was eluted from the columns with 80 µl RNase-free water (Ambion).

### One-step RT-qPCR

**qPCRBIO Probe 1-Step Go Lo-ROX (PCR Biosystems).**  Reactions were done with 5 μL RNA, 5 μL 2x qPCRBIO Probe 1-Step Go mix, 1.2 μL forward primer RdRP_SARSr-F2 (10 μM), 1.6 μL reverse primer RdRP_SARSr-R1 (10 μM), and 0.2 μL probe RdRP_SARSr-P2 (10 μM), 2 μL of 20x RTase Go, and completed with RNase-free water to 20 μL. The samples were incubated in a QuantStudio 5 (Applied Biosystems/ThermoFisher Scientific). Reverse transcription was performed for 10 minutes at 45˚C. The DNA polymerase was activated for 2 minutes at 95˚C and the samples underwent 50 cycles of denaturation (5 seconds at 95˚C) and annealing/extension (30 seconds at 60˚C). A plate read was included at the end of each extension step. Each sample was run in duplicate.

**TaqMan Fast Virus 1-Step Master Mix (Applied Biosystems).**  Reactions were performed with 5 μL RNA, 5 μL TaqMan Fast Virus 1-Step master mix, with probes and water making the 20 μL reaction. For Charité/WHO/PHE primers, 1.2 μL forward primer RdRP_SARSr-F2 (10 μM), 1.6 μL reverse primer RdRP_SARSr-R1 (10 μM), and 0.2 μL probe RdRP_SARSr-P2 (10 μM), and 7 μL RNase-free water were used. For CDC primers (EUA kit IDT), 1.5 μL of each primer-probe premixture (N1, N2 or RNAseP) and 8.5 μL water were used. The samples were run in a QuantStudio 5 (Applied Biosystems/ThermoFisher Scientific) using the "Fast" cycling mode. Reverse transcription was performed for 5 minutes at 50˚C. The reverse-transcriptase was then inactivated for 20 seconds at 95˚C and the samples underwent 50 cycles of denaturation (3 seconds at 95˚C) and annealing/extension (30 seconds at 60˚C). A plate read was included at the end of each extension step. Each sample was run in duplicate except for Fig 5C and 5D where singlets (mimicking testing) were employed.

**Luna Universal Probe One-Step RT-qPCR (NEB).**  Reactions were performed with 5 μL RNA, 10 μL 2x Luna Universal Probe One-Step reaction mix, 1 μL Luna WarmStart RT enzyme mix, 1.5 μL of each CDC primer-probe premixture (N1, N2 or RNAseP), and 2.5 μL RNase-free water. The samples were incubated in a QuantStudio 5 (Applied Biosystems/ThermoFisher Scientific) using the "Fast" cycling mode. Reverse transcription was performed for 10 minutes at 55˚C. The samples were denatured for 1 minute at 95˚C and then underwent 50 cycles of denaturation (10 seconds at 95˚C) and annealing/extension (30 seconds at 60˚C). A plate read was included at the end of each extension step. Each sample was run in duplicate except for Fig 5C where singlets (mimicking testing) were employed.

Primer and probe sequences are supplied in Supporting Information. S1 and S2 Tables in S1 File show the volume reaction and cycling conditions.

### Virus work

Original SARS-CoV-2 Strain England 2 (England 02/2020/407073) was obtained from Public Health England and 2 lineage B 1.1.7 (VOC 2 2020212/01) was kindly provided by W. Barclay (Imperial College London). The infectious virus titre was determined by plaque assay in Vero-E6 cells. Limit of detection was 40 plaque forming units (pfu)/mL. Experiments were performed n = 3.

Vero-E6 cells were kindly provided by W. Barclay (Imperial College London). Cells were maintained in complete DMEM GlutaMAX (Gibco) supplemented with 10% foetal bovine serum (FBS; Gibco), 100 U/mL penicillin and 100μg/mL streptomycin and incubated at 37˚C with 5% $CO_2$.

### Study approval

This study was approved by Guy's and St Thomas' NHS Trust, REC Ref 18/NW/0584; and Service Delivery for King's College Hospital.

## Statistical analysis

Normality was firstly assessed prior to performing either paired t-tests (parametric) or Wilcoxon matched-pairs signed rank test in Figs comparing two variables. Serial dilutions in Figs 2A, 3B and 4B were analysed employing a semi-log regression to calculate the coefficient of determination ($R^2$). Data in Fig 2C was analysed using a Shapiro-Wilk test for normality assessment prior to analysis employing ANOVA (parametric data) or Friedman test (non-parametric data).

## Results and discussion

Our pipelines are adaptable for both manual and automatic handling; we also employed heat inactivation of virus within the swabs for easier processing. We compared three RNA extraction methods, one column-based and two magnetic beads-based that can be automatized. As a benchmark, we used the QIAamp Viral RNA Mini Kit (QIAGEN) as their proprietary buffer AVL inactivates SARS-CoV-2 according to CDC guidelines. We also validated three different one-step RT-qPCR kits. We used the CDC recommended N1 and N2 primer-probe sets [15] and compared these against RdRP_SARSr primers [16]. We did not test efficiency of the reverse transcription (RT) step, as we had no access to *in vitro* transcribed RNA. For these validations we received clinical swab material from St Thomas' Hospital and King's College Hospital (London, UK) and compared our results with their diagnostics clinical pipelines. Negative swabs were diagnosed as such in the clinical laboratories (not pre-pandemic). Detailed step-to-step standard operating procedures (SOPs) can be found at https://osf.io/uebvj/.

### Comparison of RNA extraction kits

We have created a flowchart of the different processing steps and combinations in our pipeline (Fig 1), which we subsequently explain in more detail.

To test the efficiency and detection range of the CDC-recommended N1 and N2 primer-probes, we amplified serial dilutions of plasmids encoding the N SARS-CoV-2 gene (positive controls provided by Integrated DNA Technologies, IDT) using the TaqMan™ Fast Virus 1-Step Master Mix (Fig 2A), FastVirus hereafter. Good linearity could be achieved up to 10 copies of DNA molecules. Using the N1 and N2 primer-probes, we compared the efficiency of RNA recovery between the column-based QIAamp Viral RNA Mini Kit (QIAGEN, QIAmp herein) endorsed by the CDC, and two magnetic bead extraction kits: the RNAdvance Blood (now RNAdvance Viral) (Beckman hereinafter) and Mag-Bind Viral DNA/RNA 96 Kit (Omega Bio-tek, Omega herein), starting from the same material (140 μL). RNA isolation from four different coronavirus positive samples (CPS) with all three kits rendered comparable cycle thresholds (Cts) when amplified with the primer-probes N1 and N2. This was the case

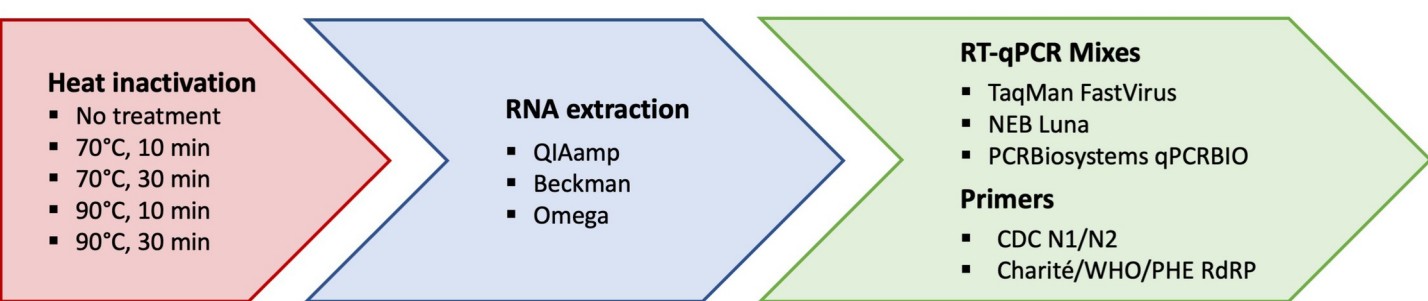

**Fig 1. Representation of our workflow.** We employed heat inactivation vs non heat inactivation [17]; compared three different RNA extraction kits (blue) followed by three RT-qPCR mixes and three sets of primers (green).

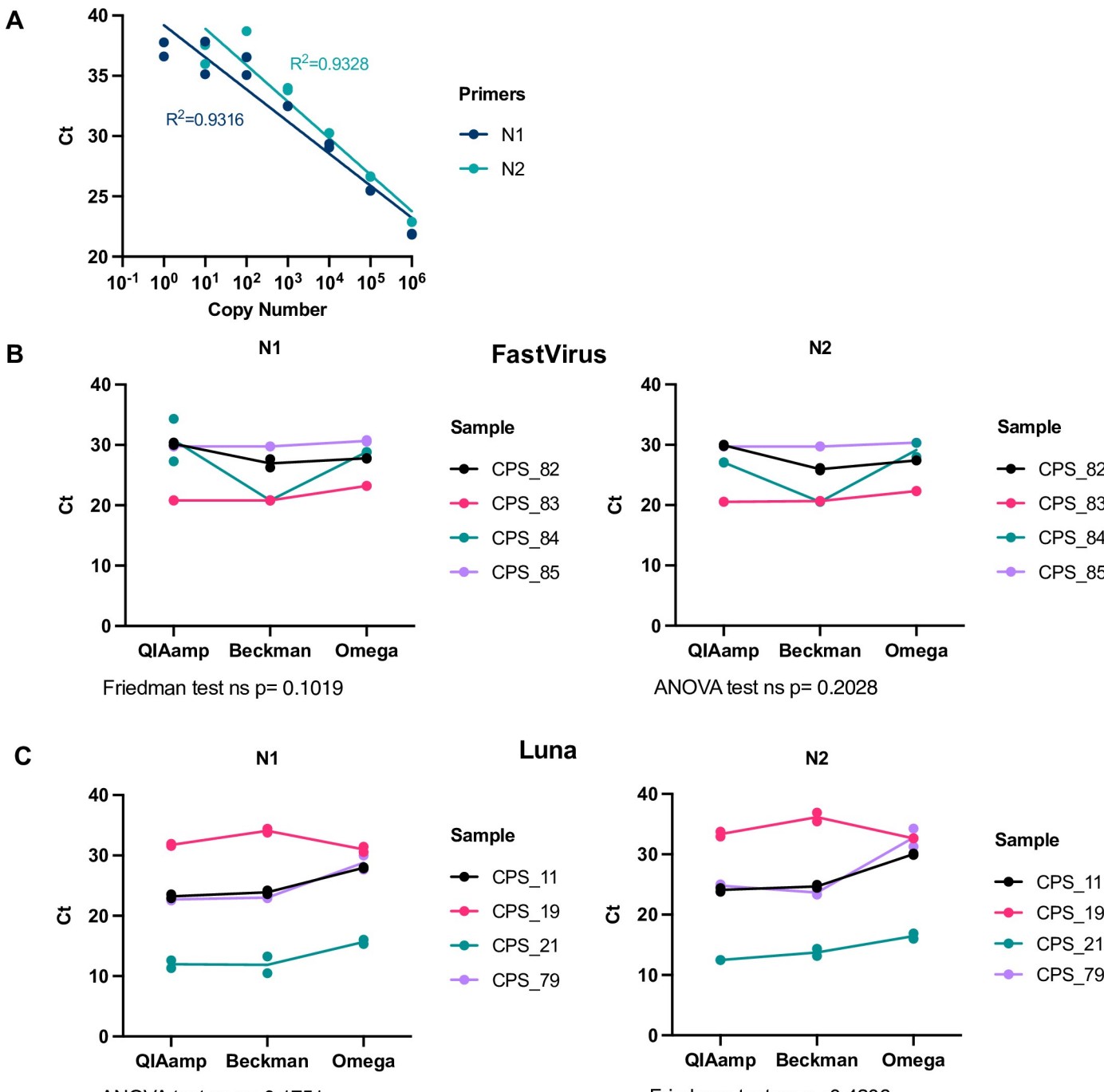

**Fig 2. Comparison between different RNA extraction methods. (A)** Dilution curve of the positive control provided by IDT (plasmid containing SARS-CoV-2 N gene) using N1 or N2 primer-probe sets with the Taqman FastVirus mix. A semi-log regression was used to calculate the coefficient of determination ($R^2$). **(B, C)** A set of four swab samples were used for RNA extraction with the indicated kit. RT-qPCR was run with N1 and N2 primer-probe sets employing the FastVirus **(B)** or the Luna Master mixes **(C)**. Shapiro-Wilk test was used for normality assessment prior to analysis employing ANOVA (parametric data) or Friedman test (non-parametric data). These samples were previously classified as positive (CPS) by the diagnostics lab, the number indicates different donors. Dots represent each individual RT-qPCR technical duplicate, line connects average of replicates and statistics were performed in average duplicates.

for two different RT-qPCR Master Mixes, FastVirus (Fig 2B) or Luna® Universal Probe One-Step RT-qPCR Kit (NEB, Luna hereafter) (Fig 2C).

## Comparison of one-step RT-qPCR kits

We also compared different one-step RT-qPCR kits to amplify swab material purified using the QIAamp viral RNA mini kit which we considered our 'benchmark' given CDC guidelines on buffer AVL inactivating SARS-CoV-2. S1 Fig in S1 File shows RNA from ten different positive donors amplified with Luna, FastVirus and qPCRBIO Probe 1-Step Go Lo-ROX (PCR Biosystems, qPCRBio). All Master Mixes detected comparable amounts of RNA using primer-probes against N1 primer-probe, with the exception of donor CPS_101 which had borderline Ct values of 38 in both FastVirus and Luna and was undetectable using qPCRBio Master Mix (ANOVA p = 0.1278). Tukey's multiple comparisons showed Luna performing better than FastVirus (p adjusted = 0.007) with no samples missed by FastVirus. Thus, three different RNA extraction kits, and three different one-step RT-qPCR kits achieve almost comparable detection of viral RNA within swab material.

As a diagnostic assay, it is paramount to be able to detect very low viral loads in swab samples. To determine the efficiency of the RT-qPCR we serially diluted the RNA from a confirmed positive swab (CPS83) isolated with each one of the three different kits used in this study. All dilutions were assessed with the N1 and N2 primer-probe amplification employing the Luna Master Mix. Fig 3A shows that the RT-qPCR reaction remained linear over a $10^5$-fold RNA dilution range. To ensure that the RNA from samples with low viral loads could be reliably extracted with each one of these extraction methods, we prepared serial dilutions of swab material from three different CPS donors in Hank's Balanced Salt Solution (HBSS) + 1% bovine serum albumin (BSA) since viral transport medium contains these only with the addition of amphotericin and gentamicin. Viral RNA was isolated from these diluted swabs with the three RNA isolation kits from Fig 2. Fig 3B shows that all three kits recovered viral RNA over a wide range of concentrations, with the N gene being reliably amplified with the N1 and N2 primer-probe sets and the Luna Master Mix. CPS21 $10^{-1}$ dilution was excluded from the r calculations. The N2 primer-probe set in the donor CPS79 extracted with Omega showed poor linearity, possibly related to initial variation in the non-diluted sample.

## Heat inactivation comparison

One major limitation for many academic and commercial laboratory settings is the lack of available CL-3 laboratory space and/or Class I MSCs required to handle/open the potentially infectious swabs. Moreover, samples with high viral load pose a risk of infection for the handler. Heat treatment of viral particles has been shown effective in inactivating SARS-CoV-2 with 70˚C 5min treatment rendering viral infectivity undetectable employing Vero E6 cells (limit of detection of TCID50 assay was 100 TCID50/mL) [18–20]. Other heat treatment protocols have also been demonstrated, with variable effects on PCR sensitivity [21]. We set out to establish the effect of heat inactivation on the sensitivity of SARS-CoV-2 detection.

We firstly assessed different temperature and time conditions for heat treatment of both SARS-CoV-2 original and B 1.1.7 variants. The novel variant of SARS-CoV-2 B.1.1.7 was originally described in December 2020 in the UK, firstly detected in samples as early as 20th September 2020 [22]. Since then, it has spread to many other countries where it is the predominant variant, together with the recent delta. We thus evaluated the effect of heat on detecting of B.1.1.7. We performed serial dilutions of cultured virus in viral transport medium, extracted RNA employing Beckman and assessed the presence of N using the N1 and N2 primer-probe combinations. Fig 4 shows that heat at either 70˚C (Fig 4A) or 90˚C (Fig 4B) for 30 minutes did not alter the detection of viral RNA (red vs blue). Fig 4C and S2A Fig in S1 File show that treatment of B.1.1.7 with 70˚C or 90˚C for 10 and 30 mins inactivates this variant of SARS-CoV-2 virus. S2B Fig in S1 File shows inactivation of the original variant at all

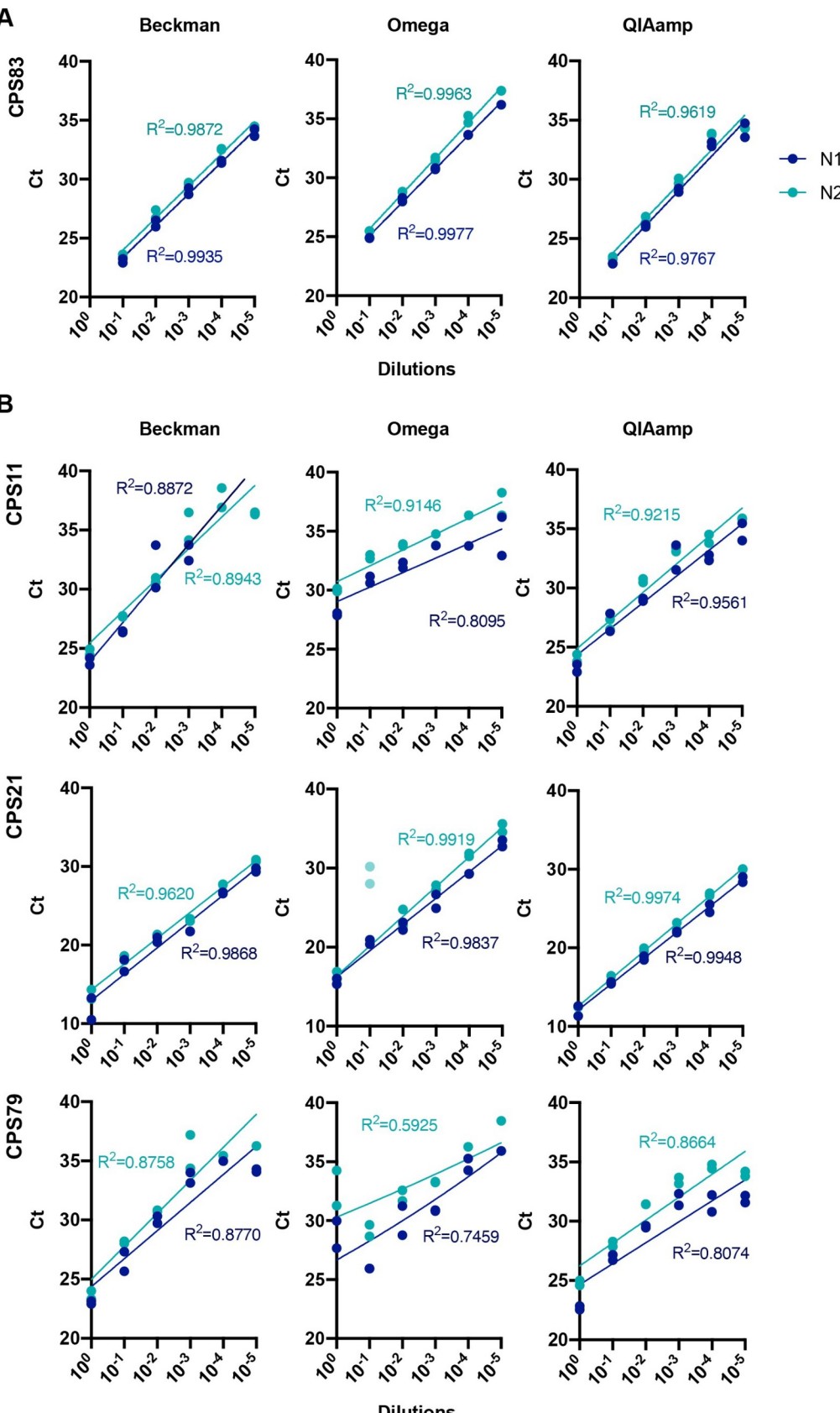

**Fig 3. Sensitivity of qPCR detection by serial dilutions of extracted RNA or swab samples. (A)** RNA from sample CPS83 was serially diluted and extracted with three different methods. RT-qPCR was run with N1 and N2 primer-probe sets with Luna Master Mix. **(B)** Three distinct positive swab samples (CPS) were serially diluted followed by RNA extraction by the indicated method. RT-qPCR was run with N1 and N2 primer-probe sets with Luna Master Mix. Dots represent each individual technical duplicate. A semi-log regression was used to calculate the coefficient of determination ($R^2$).

temperatures and times (limit of detection 40 pfu/mL). Importantly, we observed a reduction in infectivity, and not inactivation (according to our 40 pfu/mL limit), when temperatures between 60 and 70 degrees were applied for 30 min (S3 Fig in S1 File), making the use of well-calibrated thermometers or use of higher temperatures/longer time periods critical.

We then assessed if heat treating nasopharyngeal swab material could be a method of treating samples within their original unopened collection tubes without compromising RT-qPCR results. Data shown were obtained employing QIAamp RNA extraction. We first assessed heating sample aliquots at 70°C for 30min with different samples and performed RT-qPCR with N1 primer-probes and Luna Master Mix. As Fig 5A shows, we did not observe any change in Ct values upon heat treatment of the sample (t-test p = 0.1946).

We then set out to test a wider range of heat-inactivation conditions on six confirmed positive samples using two extraction methods. We treated aliquots of the same sample with no heat, 70°C for 10min, 70°C for 30min, 90°C for 10min or 90°C for 30min and extracted RNA employing the QIAamp kit. We employed a dry metallic bead bath to heat the sample tubes, since water baths are not allowed in Cat-3 laboratories due to the risk of spillage. Our results showed that none of the heat conditions altered the Ct values (ANOVA for N1 p = 0.3656 and Friedman test for N2 p = 0.3469, Fig 5B). Both the N1 and N2 primer-probe sets gave reliable and near-identical amplification of viral RNA; however, we noticed that the RdRP primer-probe set failed to amplify viral samples with high Ct values. These results of high Ct values for RdRP were also observed when we used the MagMax kit (ThermoFisher Scientific) as an extra RNA extraction method (S4 Fig in S1 File). Comparison of the different primer-probe combination for RdRP rendered similar results (S5 Fig in S1 File) and as previously shown [13].

To confirm the reproducibility of our results, we employed another distinct set of samples, assessing both positive and negative samples. We aliquoted swab material, warmed it at 70°C for 30min, extracted their RNA using QIAmp and performed RT-qPCR using Luna Master Mix using primer-probe N1. Ct values did not change upon heat inactivation as observed previously (t-test p = 0.5578). S6 Fig in S1 File shows the data for N2 and RNAseP. We detected one previously diagnosed sample as negative (sample 27) in which we could amplify N1 at a Ct of 37.489, which is at the limit of detection. All our water controls (no template and water template) yielded no amplification. RNAse P controls are in S6 Fig in S1 File.

To further test a higher temperature we employed 90°C for 10min in 93 samples, extracted their RNA using Beckman and performed RT-qPCR using FastVirus Master Mix using primer-probes for N1, N2 and RNAseP. Fig 5D shows the results for those samples where we detected positive amplification in either of the treatments (88 for N1 and 90 for N2). More samples were lost upon heating, although statistical analysis of Ct ranges remained similar between heat vs non-heat (S3 Table in S1 File). N2 primer-probe appeared more heat-resistant, as 89 samples were detected vs 88 with N1 primer-probes. Interestingly, we observed no clear trend for samples with high Ct values lost upon heat treatment (S7 Fig in S1 File).

Our work shows different workflows for nasopharyngeal swab processing for SARS-CoV-2 detection employing different combinations of inactivation, extraction and amplification. We present data for the validation of two viral RNA purification kits (Beckman and Omega) as alternatives to the QIAamp viral RNA mini kit. We have also tested three alternative,

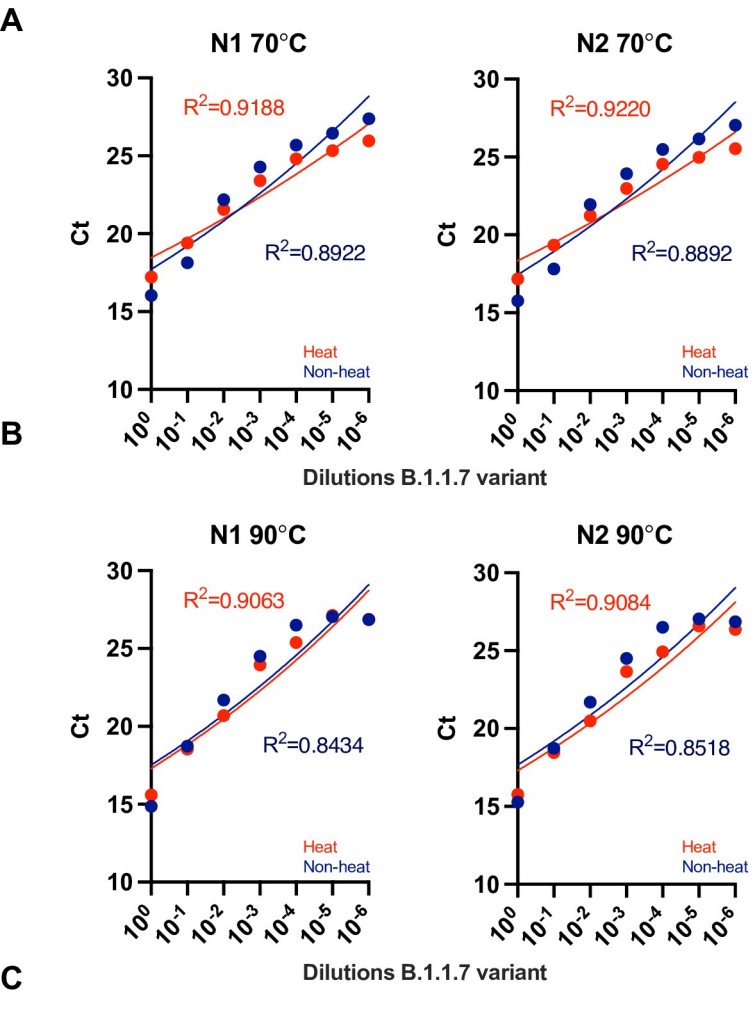

**Fig 4. Sensitivity of qPCR detection by serial dilutions of viral stocks of the new B.1.1.7 variant.** B.1.1.7 SARS-CoV-2 viral stocks were serially diluted in viral transport medium, extracted employing Beckman and assessed employing N1 and N2 primer-probe sets using the FastVirus Master Mix. Samples were heat treated for 30 minutes with either 70˚C (**A**) or 90˚C (**B**). Dots represent the mean of the qPCR technical duplicates. A semi-log regression was used to calculate the coefficient of determination ($R^2$). (**C**). Results of plaque assays for heat treatment of cultured SARS-CoV-2 B.1.1.7 variant (n = 3).

commercially available one step RT-qPCR kits (FastVirus, Luna and PCRBio) and assessed different recommended primer-probe sets (N1, N2, RdRP) which can currently detect all circulating SARS-CoV-2 variants to date.

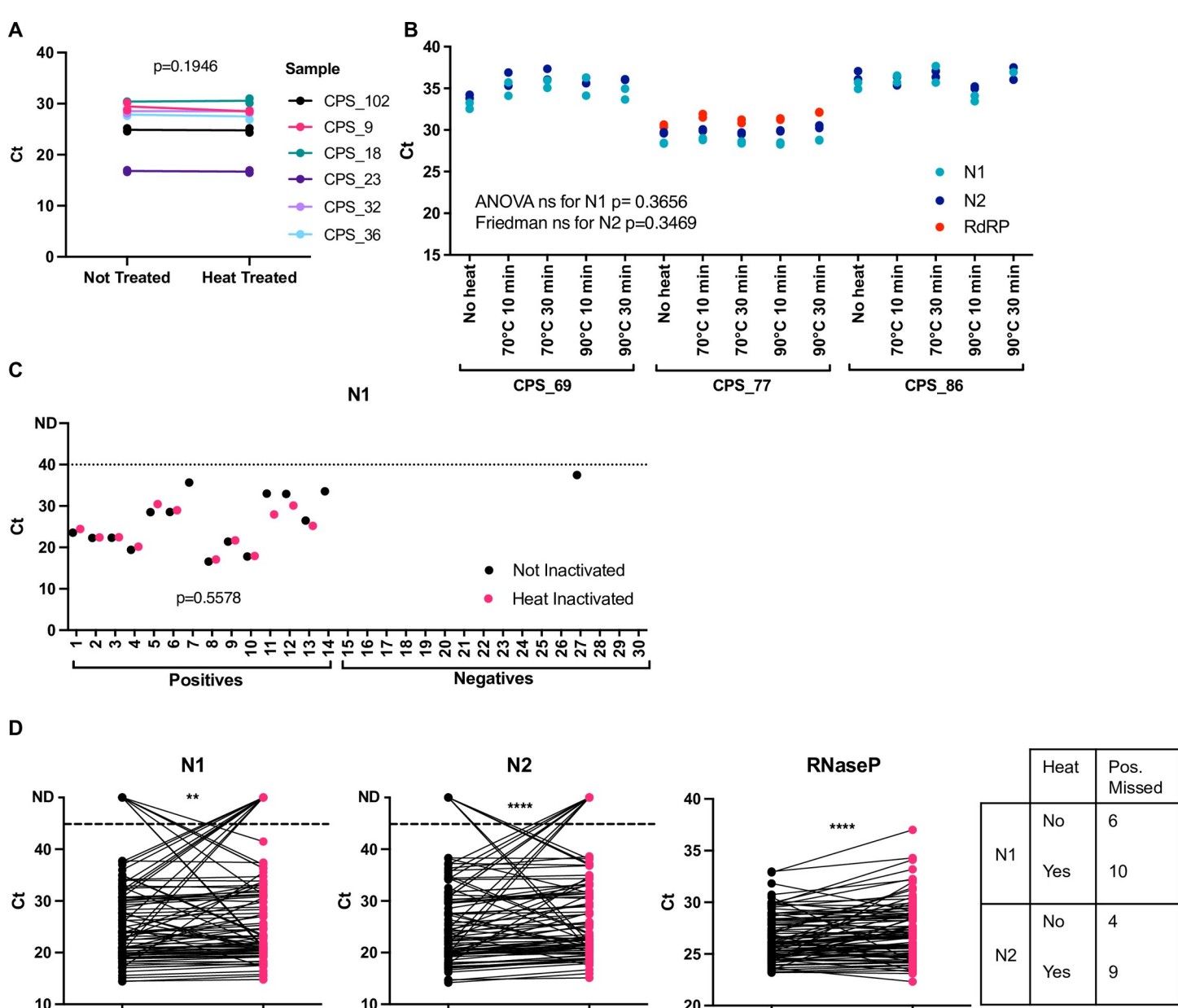

**Fig 5. Primer comparisons and heat inactivation of nasopharyngeal swab samples.** (**A**) A different set of six positive samples were used to compare directly non treated with heat treated at 70˚C for 30 min. RNA extraction was done with QIAamp and RT-qPCR with N1 primers and NEB Luna mix. (**B**) Three additional positive samples were subjected to different temperatures and incubation times as indicated, with RNA extracted by QIAamp. All three primer-probe sets from panel A were used, together with Taqman FastVirus master mix. (**C**) 30 additional samples, 14 from positive donors (1–14) and 16 from negative donors (15–30) were used to compare directly non treated with heat treated at 70˚C for 30 min. RNA extraction was done with Beckman and RT-qPCR with N1 primer-probes and Luna Master Mix. Paired t-test was employed to compare the effect of heat. (**D**) 93 additional samples were non treated or heat treated at 90˚C for 10 min. RNA extraction was done with Beckman and RT-qPCR with N1, N2 and RNAseP primer-probes and FastVirus Master Mix. Samples were run in singlets.

With regards to extraction methods, our data in Fig 2 suggest that Beckman performed better as seen for sample CPS_84. This is also supported by data in Fig 3 where CPS_21 and CPS_79 samples did not show good linearity when diluted and extracted with Omega. When analysed together, we did not observed any statistically significant difference between Luna, FastVirus and PCRBio, although one sample was not detected employing PCRBio and Luna appeared to

perform better than FastVirus (S1 Fig in S1 File). The difference in Ct values between Luna and FastVirus was very small and did not alter overall sensitivity (mean Ct difference 0.8335).

We found that N1 and N2 primer-probe are more sensitive than RdRP (Fig 5B), as reported by others [13]. This may likely be due to higher amount of sub-genomic RNAs encoding for N [23]. With regards to inactivation, most recommended viral inactivation protocols use a combination of guanidinium thiocyanate (GTC) and Triton X-100. GTC became scarce due to its wide use, it is quite toxic and not compatible with some kits, and the use of chemical inactivation protocols after sample collection inherently requires opening of the swab sample, with the consequent risk of exposure for lab staff. Our data show that heat, which is an economic and fast way of inactivating the virus, inactivates the original and B 1.1.7 variants at 70˚C or 90˚C for 10 and 30 mins (Fig 4 and S2 Fig in S1 File). Importantly, we observed that high viral concentrations using inadequate temperatures between 60 and 70˚C were not fully inactivated (S3 Fig in S1 File). We observed this in early experiments where were relied on a heat block thermometer, which was proven inaccurate and set at a temperature of ~62–63˚C instead. This highlights the need to accurately measure temperatures when performing inactivation of clinical samples–or the use of temperatures above 70˚C. Importantly, temperatures of the swabs must be considered when performing heat inactivation, since swabs kept in the fridge will take longer to reach a certain temperature vs swabs kept at room temperature. Our data also highlight that adequate titrations employing high viral loads of SARS-CoV-2 are required to establish if full inactivation has been achieved. 70˚C during 30 min appeared to have no effect on sensitivity (Fig 5C) while 90˚C during 10 min appeared to decrease sensitivity (Fig 5D). Our data supports heating at temperatures below 90˚C, method that may be used to reduce the need for CL-3 laboratory and to speed up sample processing given the chemical inactivation methods are labour-intensive increasing the risk of exposure to the lab staff. Some laboratories have implemented dry heat in ovens [4] to inactivate samples; we propose the use of dry heat with beads as it allows for both high and low throughputs and is safe against possible sample leaks (beads can be disinfected at the moment of leakage). Our pipeline can therefore be implemented in places that only have CL-2 facilities to detect SARS-CoV-2. Our results differ from those observed by others with regards to samples with low viral loads being lost upon heating [21] since samples with low viral loads were still detected and some samples were only detected upon heat treatment (S7 Fig in S1 File). Our higher temperatures employed may possibly denature RNAses and/or facilitate viral RNA denaturation while preserving enough integrity for detection with the N1 and N2 primer-probe sets. Regardless, we advise the use of 70˚C over 90˚C when possible.

Together these data highlight the need for performing cross-validations of RNA extraction kits and primer-probe pairs prior to implementing in diagnostics, with an emphasis on the need of using clinical samples (rather than diluted RNA or plasmid DNA) to establish 'real-world' data that better relate to clinical samples. Swab material and inherent inhibitors will perform variably with different workflows and we thus highlight the need to assess their performance–a task in which diagnostics laboratories can collaborate with academic institutions to speed up the establishment of new protocols. Moreover, establishing limits of detection at each laboratory purchasing international standards such as those provided by the WHO, or viral cultures as directed and established by their Local Health Authorities, is essential to ensure good practice and implementation of diagnostics.

## Conclusions

Based on the above, and understanding that including RT-qPCR duplicates may decrease the number of samples a diagnostic laboratory can process (particularly if employing 96 well plates), we suggest to:

- employ heat (70˚C preferably to 90˚C) for 10-30min. Ensure temperature is at least 70˚C;

- preferably employ N1 and N2 primer-probes vs RdRP;

- test samples without RT-qPCR technical replicates to increase the testing throughput;

- run duplicates in case of borderline ≥36Ct [24] and always check amplification curves of samples. If 1) amplification is shown reproducibly consider it a positive sample with low viral load 2) amplification unclear (one replicate positive, one negative) for these donors to be re-tested as soon as possible to confirm positive or negative detection of SARS-CoV-2 regardless of symptoms. Although we acknowledge the limitations, if possible, re-swabbing individuals with unclear or discrepant results is highly recommended as a first option.

## Supporting information

**S1 File. Supporting information contains Supporting information for materials and methods, S1-S7 Figs and S1-S3 Tables.**
(PDF)

## Acknowledgments

The authors thank the many volunteers at King's College London that were keen and offered to help during the pandemic.

## Author Contributions

**Conceptualization:** Michael H. Malim, Stuart Neil, Rocio Teresa Martinez-Nunez.

**Data curation:** Maria Jose Lista, Pedro M. Matos, Thomas J. A. Maguire, Kate Poulton, Elena Ortiz-Zapater, Robert Page, Helin Sertkaya, Aoife M. O'Byrne, Clement Bouton, Ruth E. Dickenson, Mattia Ficarelli, Esperanza Perucha, Rocio Teresa Martinez-Nunez.

**Formal analysis:** Maria Jose Lista, Pedro M. Matos, Thomas J. A. Maguire, Kate Poulton, Elena Ortiz-Zapater, Robert Page, Helin Sertkaya, Ana M. Ortega-Prieto, Mattia Ficarelli, Rocio Teresa Martinez-Nunez.

**Funding acquisition:** Katie Doores, Manu Shankar-Hari, Jonathan Edgeworth, Michael H. Malim, Stuart Neil, Rocio Teresa Martinez-Nunez.

**Investigation:** Edward Scourfield, Monica Agromayor, Esperanza Perucha, Mark Zuckerman.

**Methodology:** Maria Jose Lista, Pedro M. Matos, Thomas J. A. Maguire, Kate Poulton, Elena Ortiz-Zapater, Robert Page, Helin Sertkaya, Ana M. Ortega-Prieto, Ruth E. Dickenson, Mattia Ficarelli, Jose M. Jimenez-Guardeño, Mark Howard, Gilberto Betancor, Rui Pedro Galao, Suzanne Pickering, Adrian W. Signell, Harry Wilson, Mark Tan Kia Ik, Katie Doores, Monica Agromayor, Hannah E. Mischo.

**Project administration:** Monica Agromayor, Rocio Teresa Martinez-Nunez.

**Resources:** Penelope Cliff, Amita Patel, Eithne MacMahon, Emma Cunningham, Juan Martin-Serrano, Rahul Batra, Jonathan Edgeworth, Mark Zuckerman, Michael H. Malim, Stuart Neil.

**Supervision:** Monica Agromayor, Esperanza Perucha, Hannah E. Mischo, Rocio Teresa Martinez-Nunez.

**Visualization:** Maria Jose Lista, Pedro M. Matos, Thomas J. A. Maguire, Robert Page, Ana M. Ortega-Prieto.

**Writing – original draft:** Maria Jose Lista, Pedro M. Matos, Thomas J. A. Maguire, Kate Poulton, Elena Ortiz-Zapater, Robert Page, Helin Sertkaya, Mattia Ficarelli, Hannah E. Mischo, Rocio Teresa Martinez-Nunez.

**Writing – review & editing:** Maria Jose Lista, Pedro M. Matos, Thomas J. A. Maguire, Kate Poulton, Elena Ortiz-Zapater, Ana M. Ortega-Prieto, Ruth E. Dickenson, Jose M. Jimenez-Guardeño, Mark Howard, Rui Pedro Galao, Suzanne Pickering, Monica Agromayor, Esperanza Perucha, Hannah E. Mischo, Mark Zuckerman, Michael H. Malim, Stuart Neil, Rocio Teresa Martinez-Nunez.

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
