## [Decision Letter · Decision Letter 0]

10 Jun 2021

PONE-D-21-12761

Resilient SARS-CoV-2 diagnostics workflows including viral heat inactivation

PLOS ONE

Dear Dr. Martinez-Nunez,

Thank you for submitting your manuscript to PLOS ONE. After careful consideration, we feel that it has merit but does not fully meet PLOS ONE’s publication criteria as it currently stands. Therefore, we invite you to submit a revised version of the manuscript that addresses the points raised during the review process.

Reviewers note significant improvements in this work. However, authors need to prepare responses to new reviewers' comments and submit manuscripts in accordance with these comments.

We look forward to receiving your revised manuscript.

Kind regards,

Ruslan Kalendar, PhD

Academic Editor

PLOS ONE

Journal Requirements:

Reviewers' comments:

Reviewer's Responses to Questions

**Comments to the Author**

1. Is the manuscript technically sound, and do the data support the conclusions?

Reviewer #1: No

Reviewer #2: Yes

Reviewer #3: Yes

2. Has the statistical analysis been performed appropriately and rigorously? 

Reviewer #1: N/A

Reviewer #2: Yes

Reviewer #3: Yes

3. Have the authors made all data underlying the findings in their manuscript fully available?

Reviewer #1: Yes

Reviewer #2: Yes

Reviewer #3: Yes

4. Is the manuscript presented in an intelligible fashion and written in standard English?

Reviewer #1: Yes

Reviewer #2: Yes

Reviewer #3: Yes

5. Review Comments to the Author

Reviewer #1: 

Lista et al. pursue two main goals with this study. On the one hand they evaluate and compare different extraction methods and PCR chemistry and primersets, on the other they try to validate heat-inactivation protocols for SARS-CoV-2 in clinical samples. The title is not well chosen and does not represent what is actually done in this work. This manuscript has been reviewed before by other reviewers and revised by the authors.

I will address a few general points about this study:

1. Claims about “complete inactivation” of SARS-CoV-2 by heat inactivation in this study are not supported by the data. I am missing a lot of details in the methods about the exact methodology used for cell culture experiments, as well as the number of repeats carried out for the experiments, etc.. In general, claims of “complete inactivation” of infectivity are held to very high standards and there are good studies in the literature about other viruses that show the amount of scrutiny necessary to guarantee that a certain procedure will with certainty abolish all infectivity in samples (e.g. Smither et al. “Buffer AVL Alone Does Not Inactivate Ebola Virus in a Representative Clinical Sample Type.” JCM). There have been good studies about this for SARS-CoV-2 in the past, see e.g. Pastorino et al. “Evaluation of heating and chemical protocols for inactivating SARS-CoV-2”. Instead of declaring complete inactivation, authors should state a reduction of infectivity, in relation to their limit of detection in plaque forming units.

2. The authors assess sensitivity of different extraction workflows by serial dilution of clinical samples, with duplicate measurements, essentially performing hit-rate analysis without a quantitative reference and without giving detection limits (figure3). Additionally they perform serial dilution with plasmids as quantitative reference (figure2). All experiments should be described more in detail in the methods (incl. repeats performed).

In general, the methodology is inadequate to evaluate analytical performance of molecular methods. I would expect use of a quantitative standard (i.e. Qnostics, Acculex, etc, or WHO-Standard, which is now available for SARS-CoV-2) for serial dilutions near LoD with 8-21 repeats for each step and determination of LoD by probit analysis (log regression, 95% probability of detection)

The use of DNA-plasmids to simulate positive material of an RNA virus is several decades outdated and absolutely unacceptable. The authors state that they bought the plasmids from IDT; they could just as well have bought synthetic RNA, or done IVT on their DNA target, or bought a commercial reference standard, as stated above.

3. Why did authors choose to compare N1/N2 and Corman’s RdRp assay? It had been known very early that the Charite RdRp is a poor assay (i.e. Vogels et al.’s preprint on comparing LDTs in April 2020, later published in nature microbiology), making the comparison a foregone conclusion. Corman’s E-Sarbeco is substantially better and more commonly used for SCoV-2 detection, making it a fairer competitor, however, these kinds of experiments have already been done and published a year ago.

Authors state that the RdRp assay will not detect subgenomic RNA, however, the E-gene is also abscent in the vast majority of subgenomic RNA (Alexandersen et al. 2020, Nature communications). To avoid detecting subgenomic RNA altogether, there are other RdRp assays that perform very well like e.g. Chan et al.’s RdRp/Hel-assay (Chan et al. 2020 JCM).

4. The number of samples for workflow comparison (largely in fig5) is too small to draw valid conclusions (e.g. 10 positive and 10 negative for extraction workflow comparison). Clinical validation experiments such as this should aim at 100 samples minimum; value of the dataset increases with more samples.

5. Fig.5C raises some questions, I believe this is what Reviewer#1 was referring to in their final question. Right now, it looks as if negative samples were positive for CDC-N1 in roughly 50% of measurements. If this is the case, this experiment should be discarded and repeated after decontamination. CDC-N1 does not produce unspecific false-positives. (Note: Oligos from IDT were consistently contaminated with N1/2 and E-Sarbeco positive material from the start of the pandemic until early Fall 2020. Authors state that they buy oligos from IDT.)

6. As alluded to in previous points, the methods section should be expanded with details about analytical (fig2-4) and clinical (fig5) evaluation and cell culture experiments. Right now, it contains details about execution of individual extraction and PCR-protocols. These could be moved to supplement or replaced with “carried out according to manufacturer’s instructions” where applicable. Some of the details I am referring to are mentioned in the figure legend, but the figure legend is an addition and does not replace the methods section.

On a general note, this manuscript gives the impression of containing two largely separate projects, one of them being the inactivation aspect, and the other being analytical and clinical validation of different extraction methods. Both are executed haphazardly and strung together in a somewhat confusing fashion. When looking at the general topics, I see clear value in the comparison of different extraction workflows, as these studies are instrumental for diagnostic labs to make informed decisions about which products to employ for SARS-CoV-2 detection. The heat-inactivation topic has been published extensively in early/mid 2020 and this work does not provide anything valuable to the existing literature. My recommendation would be to restructure the manuscript and focus largely on the validation data, add more datapoints to the existing experiments and either remove the inactivation topic entirely or reducing it merely to impact on PCR performance without making any claims about efficacy of inactivation.

Reviewer #2: 

In the present study Lista et al. performs a detailed comparison of 3 commercial RNA extraction kits, 3 different RT-aPCR mixes and 3 separate target genes, for the detection of SARS-CoV-2 virus in nasopharyngeal swaps collected during the current SARS-CoV-2 pandemic. Moreover, Lista et al. assess the impact of heat-inactivation of the virus (to reduce the safety requirements) on subsequent virus detection.

In general, the revised manuscript is well written, the results are clearly presented, and the methods described in sufficient details. From their study Lista et al. shows:

• The 3 tested commonly used commercial RNA extraction kits provided similar results.

• The 3 RT-qPCR mixes tested, showed no significant differences

• Detection of the N gene was more reliable than that of RdRP (as published elsewhere)

• Heat-inactivation of the virus sample specimens have limited impact on the sensitivity of the RT-qPCR for detection of SARS-CoV-2 in clinical specimens.

The manuscript in its revised form includes the appropriate statistical tests that together with the additional results included in the revision clearly support the authors conclusions.

While mass testing for SARS-CoV-2 using large-scale kit-free procedures has been implemented in most industrialized countries, the use of small-scale kit-based protocols/work flows still may be useful in settings where access to highly advanced and automated systems are limited. As such, I find the results presented by Lista et al. an important contribution to the field.

Reviewer #3: 

The manuscript presents some open protocols for SARS-CoV-2 detection intended to help diagnostic in cases of lack of reagents or equipments, to non-experienced laboratories or to laboratories with no access to diagnostic kits. The goal is interesting and might be very useful to many laboratories around the world. Interestingly, the authors provide standard operating procedures to all their methods through a web repository. However, the manuscript might be improved at several points:

- The continuous narrative style of the Results section makes difficult to follow what is being tested in each section, adding subsection titles might be helpful.

- Authors should not use the term strain to refer to B.1.1.7 variant or to the so called original strain. Moreover, the term original strain is not very informative, was it an EU1 isolate? Was it an earlier isolate? In any case the authors are not using strains, they are using clinical isolates belonging to one or another variant.

- Why do the authors use RdRp probe and primers set from the Charité protocol and not the gene E or gene N probe and primer sets? RdRp is known to be the less reliable among the Charité set. Moreover RdRp is not used systematically, but only in the last heat-inactivation section, so probably it might be better to delete it from the work.

- The authors use RNAseP to control sample quality, but these controls are mentioned only at the end of the Results section. Authors should introduce them in the first paragraph of the Results, when they introduce the primers and probe sets.

- What is the difference between Fig. 3A and Fig. 3B? If there is no difference it might be better to include both in a single panel.

- The heat inactivation experiments are an important part of the manuscript, yet they are shown as supplementary material. The main text shows a table, Fig. 4C, summarizing the results of the experiments shown in Supp. Fig. 2A, but in Fig. 4C the reader cannot appreciate the difficulties in evaluating Supp. Fig. 2A. The difference in the native controls between the original variant and the B.1.1.7 variant is huge. The authors state in the figure legend that B.1.1.7 plaques are smaller, but the fact is that plaques are seen only in the -1 dilution. Given the critical value of these data to assess safety more convincing images should be provided.

- First paragraph in page 5 are not results. Better move it to discussion section.

- In Fig. 5 the authors state that extractions in these experiments were done by QIAamp, this means that all the samples had been inactivated with AVL buffer and this should be stated in a more explicit manner in the text and the figure legend, otherwise the labels "Not inactivated" and "Heat Inactivated" may be misleading because all the samples treated with AVL are inactivated.

6. PLOS authors have the option to publish the peer review history of their article (what does this mean?). If published, this will include your full peer review and any attached files.

Reviewer #1: **Yes: **Dominik Nörz

Reviewer #2: No

Reviewer #3: No

---

## [Author Response · Author response to Decision Letter 0]

27 Jul 2021

Point by Point answers

Reviewer #1:

Lista et al. pursue two main goals with this study. On the one hand they evaluate and compare different extraction methods and PCR chemistry and primersets, on the other they try to validate heat-inactivation protocols for SARS-CoV-2 in clinical samples. The title is not well chosen and does not represent what is actually done in this work. This manuscript has been reviewed before by other reviewers and revised by the authors.

I will address a few general points about this study: 

1. Claims about “complete inactivation” of SARS-CoV-2 by heat inactivation in this study are not supported by the data. I am missing a lot of details in the methods about the exact methodology used for cell culture experiments, as well as the number of repeats carried out for the experiments, etc.. In general, claims of “complete inactivation” of infectivity are held to very high standards and there are good studies in the literature about other viruses that show the amount of scrutiny necessary to guarantee that a certain procedure will with certainty abolish all infectivity in samples (e.g. Smither et al. “Buffer AVL Alone Does Not Inactivate Ebola Virus in a Representative Clinical Sample Type.” JCM). There have been good studies about this for SARS-CoV-2 in the past, see e.g. Pastorino et al. “Evaluation of heating and chemical protocols for inactivating SARS-CoV-2”. Instead of declaring complete inactivation, authors should state a reduction of infectivity, in relation to their limit of detection in plaque forming units.

Thank you for the helpful suggestion, we agree with the reviewer and have included the limit of detection of our plaque assays in the Figure. We performed our assays n=3 times and we have now included that information too. 

2. The authors assess sensitivity of different extraction workflows by serial dilution of clinical samples, with duplicate measurements, essentially performing hit-rate analysis without a quantitative reference and without giving detection limits (figure3). Additionally they perform serial dilution with plasmids as quantitative reference (figure2). All experiments should be described more in detail in the methods (incl. repeats performed). In general, the methodology is inadequate to evaluate analytical performance of molecular methods. I would expect use of a quantitative standard (i.e. Qnostics, Acculex, etc, or WHO-Standard, which is now available for SARS-CoV-2) for serial dilutions near LoD with 8-21 repeats for each step and determination of LoD by probit analysis (log regression, 95% probability of detection). The use of DNA-plasmids to simulate positive material of an RNA virus is several decades outdated and absolutely unacceptable. The authors state that they bought the plasmids from IDT; they could just as well have bought synthetic RNA, or done IVT on their DNA target, or bought a commercial reference standard, as stated above.

We performed our quantitative serial dilutions employing cultured virus (Figure 4) to assess linearity, in addition to showing dilution of clinical swabs which we agree reflect much better the quality of the data vs plasmids. While we appreciate that commercial supplies can be purchased to this end, there are examples in the literature that show linearity of performance using cultured virus [1]. We have included a phrase to make the reader aware in our Discussion: ‘Moreover, establishing limits of detection at each laboratory purchasing international standards such as those provided by the WHO, or viral cultures as directed and established by their Local Health Authorities, is essential to ensure good practice and implementation of diagnostics’. Our manuscript aims to determine resilient protocols and thus to provide guidance and data performance on reagents/protocols that are available in molecular laboratories. Testing laboratories may decide in accordance with the Local Health Authority to establish their limit of detection employing either in house viral cultures or international standards, which are required regardless of what a manufacturer of IVD CE products may provide as evidence. Beyond these comments and suggestions, we believe performing these experiments is out of the scope of our work. 

3. Why did authors choose to compare N1/N2 and Corman’s RdRp assay? It had been known very early that the Charite RdRp is a poor assay (i.e. Vogels et al.’s preprint on comparing LDTs in April 2020, later published in nature microbiology), making the comparison a foregone conclusion. Corman’s E-Sarbeco is substantially better and more commonly used for SCoV-2 detection, making it a fairer competitor, however, these kinds of experiments have already been done and published a year ago. Authors state that the RdRp assay will not detect subgenomic RNA, however, the E-gene is also abscent in the vast majority of subgenomic RNA (Alexandersen et al. 2020, Nature communications). To avoid detecting subgenomic RNA altogether, there are other RdRp assays that perform very well like e.g. Chan et al.’s RdRp/Hel-assay (Chan et al. 2020 JCM).

We chose RdRp as it is still employed in several commercially available kits such as Primer Design or Viasure. We believe that adding more data to current literature does not harm our manuscript; if anything it shows reproducibility between laboratories, an essential task in diagnostics. With regards to detection of genomic/subgenomic RNA the implications for clinical management are not yet established, to our knowledge, and thus we only added the phrase as a comment.

4. The number of samples for workflow comparison (largely in fig5) is too small to draw valid conclusions (e.g. 10 positive and 10 negative for extraction workflow comparison). Clinical validation experiments such as this should aim at 100 samples minimum; value of the dataset increases with more samples.

We agree that more samples increase the power of the study and this is why in Figure 5D we provide nearly 100 samples as comparison. We have also modified Figure 5C including 14 positives and 16 negatives. The extraction workflow comparison was done employing serial dilutions to demonstrate the linearity of the extraction process, which we believe is the right approach and already shows differences when comparing extraction methods in our dataset.

5. Fig.5C raises some questions, I believe this is what Reviewer#1 was referring to in their final question. Right now, it looks as if negative samples were positive for CDC-N1 in roughly 50% of measurements. If this is the case, this experiment should be discarded and repeated after decontamination. CDC-N1 does not produce unspecific false-positives. (Note: Oligos from IDT were consistently contaminated with N1/2 and E-Sarbeco positive material from the start of the pandemic until early Fall 2020. Authors state that they buy oligos from IDT.)

Thank you for this observation. We can demonstrate that we did not have contamination (water negative controls as explained in our previous Supplemental Figure 6). We agree with the Reviewer that this panel added confusion; our intention was to present all our data as we obtained it. The Ct threshold that laboratories employ depends on the manufacturer and internal validations. In our case, we presented detection and Ct value, which was in most cases very high and thus may be deemed as negative in some settings. Having said this, we have repeated the comparison in a new Figure 5C and Supplemental Figure 6 with new samples, including 14 positives and 16 negatives.

6. As alluded to in previous points, the methods section should be expanded with details about analytical (fig2-4) and clinical (fig5) evaluation and cell culture experiments. Right now, it contains details about execution of individual extraction and PCR-protocols. These could be moved to supplement or replaced with “carried out according to manufacturer’s instructions” where applicable. Some of the details I am referring to are mentioned in the figure legend, but the figure legend is an addition and does not replace the methods section.

Thank you for this observation, we have now expanded the methods section as helpfully pointed out by the Reviewer. We preferred to leave the methods section with the detailed information as we provide certain tips about each protocol, which we have also uploaded as Standard Operating Procedures in the Open Science Framework for everyone’s access.

On a general note, this manuscript gives the impression of containing two largely separate projects, one of them being the inactivation aspect, and the other being analytical and clinical validation of different extraction methods. Both are executed haphazardly and strung together in a somewhat confusing fashion. When looking at the general topics, I see clear value in the comparison of different extraction workflows, as these studies are instrumental for diagnostic labs to make informed decisions about which products to employ for SARS-CoV-2 detection. The heat-inactivation topic has been published extensively in early/mid 2020 and this work does not provide anything valuable to the existing literature. My recommendation would be to restructure the manuscript and focus largely on the validation data, add more datapoints to the existing experiments and either remove the inactivation topic entirely or reducing it merely to impact on PCR performance without making any claims about efficacy of inactivation.

We thank the reviewer for the thoughtful comments and suggestions on the structure of our manuscript. We believe that our protocols add value to the current needs of increasing testing capacity in light of the rapid expansion of new variants such as the delta variant now sweeping the UK and other countries. We believe that our heat inactivation methods add value to existing literature as we provide details about how to perform them and specific temperatures and timings, with validation about specificity and sensitivity. We have provided new data on Figure 5C and modified the text to correctly point out inactivation according to limit of detection, as well as new sub-headings which we believe help reading and structuring our manuscript. Our aim is to provide a plethora of different protocols for the users to combine the steps that best suit their setting and needs.

Reviewer #2:

In the present study Lista et al. performs a detailed comparison of 3 commercial RNA extraction kits, 3 different RT-aPCR mixes and 3 separate target genes, for the detection of SARS-CoV-2 virus in nasopharyngeal swaps collected during the current SARS-CoV-2 pandemic. Moreover, Lista et al. assess the impact of heat-inactivation of the virus (to reduce the safety requirements) on subsequent virus detection.

In general, the revised manuscript is well written, the results are clearly presented, and the methods described in sufficient details. From their study Lista et al. shows:

• The 3 tested commonly used commercial RNA extraction kits provided similar results.

• The 3 RT-qPCR mixes tested, showed no significant differences

• Detection of the N gene was more reliable than that of RdRP (as published elsewhere)

• Heat-inactivation of the virus sample specimens have limited impact on the sensitivity of the RT-qPCR for detection of SARS-CoV-2 in clinical specimens.

The manuscript in its revised form includes the appropriate statistical tests that together with the additional results included in the revision clearly support the authors conclusions. While mass testing for SARS-CoV-2 using large-scale kit-free procedures has been implemented in most industrialized countries, the use of small-scale kit-based protocols/work flows still may be useful in settings where access to highly advanced and automated systems are limited. As such, I find the results presented by Lista et al. an important contribution to the field.

We are very grateful to the Reviewer for their comments on our manuscript and thrilled they deem it appropriate and of value to the field – this is certainly our intention when we performed all experiments. 

Reviewer #3: 

The manuscript presents some open protocols for SARS-CoV-2 detection intended to help diagnostic in cases of lack of reagents or equipments, to non-experienced laboratories or to laboratories with no access to diagnostic kits. The goal is interesting and might be very useful to many laboratories around the world. Interestingly, the authors provide standard operating procedures to all their methods through a web repository. However, the manuscript might be improved at several points:

- The continuous narrative style of the Results section makes difficult to follow what is being tested in each section, adding subsection titles might be helpful.

We thank the Reviewer and have now added subtitles as suggested which we think help follow the manuscript

- Authors should not use the term strain to refer to B.1.1.7 variant or to the so called original strain. Moreover, the term original strain is not very informative, was it an EU1 isolate? Was it an earlier isolate? In any case the authors are not using strains, they are using clinical isolates belonging to one or another variant.

We apologise for the mistake; indeed these are different variants and come from different clinical isolates. We have now corrected ‘strain’ with ‘variant’. Thank you for spotting this.

- Why do the authors use RdRp probe and primers set from the Charité protocol and not the gene E or gene N probe and primer sets? RdRp is known to be the less reliable among the Charité set. Moreover RdRp is not used systematically, but only in the last heat-inactivation section, so probably it might be better to delete it from the work.

We tested the RdRp probe and primers to establish their performance and compare them to the more commonly used CDC N probes and primers. There are some commercial kits that still employ RdRp and, in line with the work of others (Vogels et al., 2020), we observe a decrease in sensitivity which we think important to report. 

- The authors use RNAseP to control sample quality, but these controls are mentioned only at the end of the Results section. Authors should introduce them in the first paragraph of the Results, when they introduce the primers and probe sets.

We thank the reviewer for the suggestion; as we only present the data in the last Figure we have decided to leave the explanation there.

- What is the difference between Fig. 3A and Fig. 3B? If there is no difference it might be better to include both in a single panel.

Thank you for asking this, as we realised we did not explain this well in the text, which we have now hopefully clarified. Figure 3A employed one donor’s RNA that was extracted with different kits and serially diluted. Figure 3B shows original material from 3 donors serially diluted in viral transport medium, extracted and then assessed by RT-qPCR. The latter aims to establish linearity about the extraction methods.

- The heat inactivation experiments are an important part of the manuscript, yet they are shown as supplementary material. The main text shows a table, Fig. 4C, summarizing the results of the experiments shown in Supp. Fig. 2A, but in Fig. 4C the reader cannot appreciate the difficulties in evaluating Supp. Fig. 2A. The difference in the native controls between the original variant and the B.1.1.7 variant is huge. The authors state in the figure legend that B.1.1.7 plaques are smaller, but the fact is that plaques are seen only in the -1 dilution. Given the critical value of these data to assess safety more convincing images should be provided.

We thank the Reviewer for considering this an important part of our manuscript. Our B.1.1.7 experiments have an inherent limitation since the viral stocks available had lower pfu/mL than the original UK02 variant. Moreover, B.1.1.7 has less fitness in cultured Vero E6 as compared with UK02 – this has previously been observed by Prof Wendy Barclay’s group in their pre-print work [2] and shows smaller plaque assays, which we demonstrate down to dilution -2. 

- First paragraph in page 5 are not results. Better move it to discussion section.

We thank the Reviewer for the suggestion. We have now moved the central part of this paragraph to discussion and only left a small section to introduce our datasets, which we believe now reads better.

- In Fig. 5 the authors state that extractions in these experiments were done by QIAamp, this means that all the samples had been inactivated with AVL buffer and this should be stated in a more explicit manner in the text and the figure legend, otherwise the labels "Not inactivated" and "Heat Inactivated" may be misleading because all the samples treated with AVL are inactivated.

We thank the reviewer for this observation and agree- we have now modified the labels to ‘Not Treated’ and ‘Heat Treated’.

References

1. Nörz D, Frontzek A, Eigner U, Oestereich L, Wichmann D, Kluge S, et al. Pushing beyond specifications: Evaluation of linearity and clinical performance of the cobas 6800/8800 SARS-CoV-2 RT-PCR assay for reliable quantification in blood and other materials outside recommendations. Journal of Clinical Virology. 2020;132:104650. doi: https://doi.org/10.1016/j.jcv.2020.104650.

2. Brown JC, Goldhill DH, Zhou J, Peacock TP, Frise R, Goonawardane N, et al. Increased transmission of SARS-CoV-2 lineage B.1.1.7 (VOC 2020212/01) is not accounted for by a replicative advantage in primary airway cells or antibody escape. bioRxiv. 2021. doi: https://doi.org/10.1101/2021.02.24.432576

---

## [Decision Letter · Decision Letter 1]

17 Aug 2021

Resilient SARS-CoV-2 diagnostics workflows including viral heat inactivation

PONE-D-21-12761R1

Dear Dr. Martinez-Nunez,

We’re pleased to inform you that your manuscript has been judged scientifically suitable for publication and will be formally accepted for publication once it meets all outstanding technical requirements.

Kind regards,

Ruslan Kalendar

Academic Editor

PLOS ONE

---

## [Editor Report · Acceptance letter]

25 Aug 2021

PONE-D-21-12761R1 

Resilient SARS-CoV-2 diagnostics workflows including viral heat inactivation 

Dear Dr. Martinez-Nunez:

I'm pleased to inform you that your manuscript has been deemed suitable for publication in PLOS ONE. Congratulations! Your manuscript is now with our production department. 

Kind regards, 

on behalf of

Professor Ruslan Kalendar 

Academic Editor

PLOS ONE